# Rethinking Oncologic Treatment Strategies with Interleukin-2

**DOI:** 10.3390/cells12091316

**Published:** 2023-05-05

**Authors:** Brian Ko, Naoko Takebe, Omozusi Andrews, Monish Ram Makena, Alice P. Chen

**Affiliations:** Division of Cancer Treatment & Diagnosis, National Cancer Institute, Bethesda, MD 20892, USA

**Keywords:** cytokines, IL-2, immunotherapy, nemvaleukin alfa, cancer

## Abstract

High-dose recombinant human IL-2 (rhIL-2, aldesleukin) emerged as an important treatment option for selected patients with metastatic melanoma and metastatic renal cell carcinoma, producing durable and long-lasting antitumor responses in a small fraction of patients and heralding the potential of cancer immunotherapy. However, the adoption of high-dose rhIL-2 has been restricted by its severe treatment-related adverse event (TRAE) profile, which necessitates highly experienced clinical providers familiar with rhIL-2 administration and readily accessible critical care medicine support. Given the comparatively wide-ranging successes of immune checkpoint inhibitors and chimeric antigen receptor T cell therapies, there have been concerted efforts to significantly improve the efficacy and toxicities of IL-2-based immunotherapeutic approaches. In this review, we highlight novel drug development strategies, including biochemical modifications and engineered IL-2 variants, to expand the narrow therapeutic window of IL-2 by leveraging downstream activation of the IL-2 receptor to selectively expand anti-tumor CD8-positive T cells and natural killer cells. These modified IL-2 cytokines improve single-agent activity in solid tumor malignancies beyond the established United States Food and Drug Administration (FDA) indications of metastatic melanoma and renal cell carcinoma, and may also be safer in rational combinations with established treatment modalities, including anti-PD-(L)1 and anti-CTLA-4 immunotherapy, chemotherapies, and targeted therapy approaches.

## 1. Introduction

Cytokines are low-molecular weight (less than 80 kDa) soluble proteins secreted by diverse immune cells that regulate cell signaling, and their dysregulation can lead to immunopathology and cancer [1,2]. Cytokines bind to specific cell surface receptors and regulate several genes and their associated transcription factors, and they are categorized based on structural homology into chemokines, colony-stimulating-factors (CSF), interferons, interleukins, transforming growth factors (TGF), and tumor necrosis factors [3,4]. Interleukins play a heterogeneous and complex role in carcinogenesis, cancer growth and progression, immune evasion, cancer maintenance, immunoediting, and eradication, and differential serum levels of interleukins can be indicative of varying levels of immune activation in patients [5,6,7]. To date, 40 distinct interleukins have been identified, which modulate diverse clinical entities including allergic reactions, autoimmunity, arthritis, and cancer [8]. The therapeutic targeting of interleukins by either neutralizing or bolstering their specific bioactivities may lead to new modalities of cancer immunotherapy. In this review, we describe approaches of targeting IL-2 for more effective oncologic treatment.

## 2. Interleukin-2 (IL-2)

Since the discovery of interleukin-2 (IL-2) in 1976 as a potent growth factor promoting the expansion and maintenance of function of human T cells, there has been great interest in harnessing its pleotropic activities against cancer. However, IL-2 plays a complex double-edged role in immune system activation by interfacing with diverse cell populations, exerting variable and opposing effects that can either stimulate or dampen downstream immune responses. To harness the anti-tumor potential of IL-2 more effectively and expand its therapeutic window, the balance will have to shift towards the activation of anti-tumor immune cells while minimizing the expansion of immunosuppressive immune cell populations and their associated side effects by using novel strategies including fusion and pegylated proteins, as described below.

IL-2, a 15.5-kDa cytokine, was initially discovered in the supernatant of phytohemagglutinin-activated human peripheral blood leukocytes [9]. IL-2 promotes T cell clonal expansion, drives the differentiation of naïve CD8+ T cells into effector memory and terminal effector cells, and maintains populations of CD4+ T cells. In addition to its initially recognized role in T cell proliferation, IL-2 also increases the cytolytic activity of lymphokine-activated killer cells and NK cells [10]. In activated T and NK cells, IL-2 upregulates granzyme B, perforin, and cytokine production. IL-2 plays an essential role in generating T helper 9 (Th9) cells, primes cells for Th1 and Th2 cell differentiation, and inhibits the differentiation of Th17 and T follicular helper (TFH) cells. However, IL-2 also simultaneously dampens immune responses by promoting the development and maintenance of immunosuppressive T regulatory (T_reg_) cell populations, mainly attributed to CD4+CD25+Foxp3+ T_reg_ cells [11].

The binding of IL-2 to its receptor (IL-2R) leads to the activation of key signaling pathways including JAK/STAT, PI3K/AKT, and MAPK [12]. IL-2R consists of α (CD25), β (CD122), and γ (CD132) subunits. IL-2 binds with low affinity (Kd ~10^−8^ M) to IL-2Rα, with intermediate affinity (Kd ~10^−9^ M) to IL-2Rβ and IL-2Rγ, and with high affinity (Kd ~10^−11^ M) to heterotrimeric receptors containing IL-2Rα, IL-2Rβ, and IL-2Rγ. IL-2Rα is mainly expressed by T_reg_ cells, activated CD4+ and CD8+ T cells, B cells, CD56-high NK cells, endothelial cells, and mature dendritic cells. IL-2Rβ and IL-2Rγ are predominantly expressed by T_reg_ cells, memory CD8+ T cells, monocytes, NK cells, and neutrophils [11,12,13]. The co-expression of all three subunits is needed for high-affinity IL-2 binding, and Tregs mainly express the high-affinity trimeric IL-2 receptor, whereas CD8+ T cells and NK cells predominantly express the intermediate-affinity dimeric IL-2 receptor [12] (Figure 1A).

## 3. Targeting IL-2 as Cancer Immunotherapy

The first clinical demonstration of the anti-tumor potential of high-dose intravenous IL-2 occurred in 1984 at the National Cancer Institute (NCI) under the direction of Dr. Steven Rosenberg when a 33-year-old woman with metastatic melanoma and extensive subcutaneous metastatic deposits underwent a dramatic inpatient treatment course at the National Institutes of Health Clinical Center. The administration of high-dose IL-2 produced life-threatening capillary leak syndrome and 22 lb of fluid weight gain, and the patient required temporary intubation and mechanical ventilatory support for adequate oxygenation. After weathering these severe toxicities, she was discharged from the hospital and went on to have complete disappearance of her metastatic melanoma over the next few months. Despite the severe toxicity profile, this case represented a landmark observation with the durable eradication of human cancer via a purely immunologic intervention [14,15].

In 1985, 25 patients diagnosed with metastatic cancer refractory to standard therapies were treated at the NCI with increasing doses of IL-2, until toxicity precluded dose escalation. Four out of seven patients with metastatic melanoma and three out of three patients with metastatic renal cancer had regression of their disease [14,16]. In the late 1990s, Rosenberg and colleagues reported the durability of anti-tumor responses in cancer patients treated with IL-2. Four-hundred and nine patients with metastatic melanoma or renal cancers were treated with a high dose of IL-2 (720,000 IU/kg) between September 1985 and November 1996 with a median potential follow-up of 7.1 years. Three-hundred and ninety-seven enrolled patients had undergone previous oncologic therapy. Thirty-three (8.1%) patients diagnosed with renal cell carcinoma or melanoma achieved complete response and 27 of the 33 (82%) remained in complete response from 39 to 148 months after treatment, and tumor regression was observed in all organ sites [16,17]. In total, this study demonstrated a 15% objective response rate (ORR) in 182 patients with metastatic melanoma and a 19% ORR in 227 patients with metastatic renal cancer.

The observed anti-tumor effects of IL-2 led to many studies using either a high-dose bolus regimen or a continuous infusion of IL-2 in patients with metastatic cancer [16]. Patients with metastatic melanoma or metastatic renal cell carcinoma were the most responsive to high-dose IL-2 administration, while only rare objective responses were observed in patients with other tumor types [16]. High-dose IL-2 therapy was approved by the FDA for the treatment of metastatic renal cell carcinoma in 1992 and for metastatic melanoma in 1998 [18].

## 4. Limitations of IL-2 Immunotherapy

Recombinant IL-2 has a half-life of only 5 to 7 min when administered intravenously. To achieve optimal immune-modulatory effects, high-dose IL-2 is needed, resulting in higher peak serum concentrations, and associated severe toxicities in patients. The most common side effects of IL-2 therapy are asthenia, fever, influenza-like syndrome, nausea, and vomiting. Potentially life-threatening clinical toxicities include capillary leak syndrome, which is characterized by vascular tone loss and extravasation of plasma proteins and fluid into the extravascular space with resultant hypotension and reduced organ perfusion, and pulmonary edema arising from the activation of high-affinity IL-2 receptors on lung endothelial cells, which can lead to acute hypoxemic respiratory failure. These severe adverse events (AEs) have strictly restricted the use of high-dose recombinant human interleukin-2 (rhIL-2) to intensive care settings with appropriate critical care medicine staffing and equipment [19]. To address these toxicity limitations, new therapies are being designed to circumvent the need for continuous high IL-2 dosing.

Patients who had no anti-tumor response to high-dose IL-2 had significantly greater expansion of T_reg_ cells compared to patients with objective anti-tumor responses, suggesting an association of the inhibitory role of these cells with poor clinical response (Figure 1A) [20,21,22]. To overcome these limitations, novel strategies to selectively enhance immune activation while avoiding concomitant augmentation of immune suppressor cells are needed.

## 5. Nemvaleukin Alfa

Nemvaleukin alfa (ALKS 4230; Alkermes) is an engineered IL-2-IL-2Rα fusion protein that selectively binds to intermediate-affinity IL-2 receptor complexes rather than high-affinity IL-2 receptor complexes present on the surface of T_regs_ and endothelial cells. Nemvaleukin alfa preferentially activates CD8-positive cytotoxic T cells and NK cells and reduces the expansion of immunosuppressive T_regs_, potentially enhancing the anti-cancer immune effects that are primarily mediated by CD8-positive T cells [23,24]. Unlike recombinant IL-2, nemvaleukin alfa is stable in systemic circulation, is not degraded to native IL-2, and is intrinsically active without antecedent metabolic or proteolytic activation to exert its drug–target effects [24] (Figure 1B).

The steric restriction of the binding of nemvaleukin alfa to high-affinity IL-2 receptor complexes is hypothesized to reduce certain serious and life-threatening immune-mediated toxicities of high-dose recombinant IL-2 directly related to the activation of high-affinity IL-2 receptor complexes. The high-affinity IL-2 receptor complexes expressed on T_regs_ have been shown to be a contributing factor in higher tumor infiltration of T_regs_, which correlated with worse disease prognosis.

Nemvaleukin alfa has demonstrated single-agent anti-tumor activity with both intravenous (IV) and subcutaneous (SC) routes of administration in murine and non-human primate in vivo models [24,25]. In NOD-scid IL-2Rγnull mice injected with human melanoma tumor cells, nemvaleukin alfa SC induced significantly higher counts of CD8-positive T cells and NK cells and lower counts of T_regs_ within their spleens compared to rhIL-2 treatment; evaluation of the implanted melanoma tumors revealed that nemvaleukin alfa SC increased the percentages of intratumoral T cells and CD8-positive T cells and reduced the percentages of intratumoral T_regs_ compared to the vehicle. In a B16-F10 murine melanoma cell line lung tumor metastasis model using female B6D2F1 mice, nemvaleukin alfa SC demonstrated better anti-tumor efficacy with fewer metastatic colonies per lung compared to rhIL-2 SC treatment [23].

## 6. Nemvaleukin Alfa Clinical Data

The clinical development of nemvaleukin alfa has encompassed a wide variety of advanced solid tumor malignancies, as presented in Table 1. This promising fusion protein therapy is currently undergoing evaluation in a phase 3 trial (ARTISTRY-7, NCT05092360) in patients with platinum-resistant epithelial ovarian, fallopian tube, or primary peritoneal cancer. Accrual to earlier-phase trials included patients with tumor types in which rhIL-2 and anti-PD-(L)1 agents have demonstrated substantial overall survival benefit (i.e., cutaneous melanoma, renal cell carcinoma) and those in which single-agent immune checkpoint therapy approaches have not yielded meaningful response rates (i.e., platinum-resistant ovarian cancer, pancreas adenocarcinoma). The safety and efficacy of IV or SC nemvaleukin alfa are both being evaluated across various clinical trials (Table 1), and both IV and SC administration of nemvaleukin alfa increases the expansion of CD8-positive T cells and NK cells in human subjects in a dose-dependent manner [26,27]. 

The phase 1 and 2 ARTISTRY-1 trial (NCT02799095) evaluated nemvaleukin alfa in patients with advanced solid tumors in three parts. Part A consisted of dose escalation up to 10 mcg/kg/day IV (day 1 through day 5 in 21-day cycles), Part B involved IV nemvaleukin monotherapy dose expansion in melanoma and renal cell carcinoma cohorts, and Part C evaluated the combination of nemvaleukin at 3 or 6 mcg/kg/day and pembrolizumab in 21-day cycles in patient cohorts of distinct tumor types based on previously established FDA indications for anti-PD-(L)1 agents. Intravenous nemvaleukin alfa monotherapy was found to have anti-tumor activity with objective responses in patients with mucosal melanoma, cutaneous melanoma, and renal cell carcinoma, leading to the FDA granting nemvaleukin alfa both orphan drug designation and fast track designation for the treatment of mucosal melanoma in 2021, which clearly distinguishes nemvaleukin alfa from the limited observed efficacy of aldesleukin in mucosal melanoma. For the Part B monotherapy arm, 4 patients achieved a partial response (PR), including 2 out of 30 (6.7%, 1 unconfirmed) patients with cutaneous melanoma and 2 out of 6 (33%) patients with mucosal melanoma. Twenty-one patients with melanoma achieved stable disease (SD). In the Part C combination treatment arm, 2 out of 14 (14.3%) patients with platinum-resistant ovarian cancer achieved a complete response (CR) and 2 out of 14 (14.3%, 1 unconfirmed) had PR, while six other patients had stable disease (SD). Partial responses were also observed in patients with Hodgkin lymphoma, triple-negative breast, pancreas, esophagus, cervix, bladder, head and neck, and lung cancer. These early signals of anti-tumor effects in multiple human cancer types outside of cutaneous melanoma and renal cell carcinoma, which have not been consistently observed with aldesleukin or various pegylated IL-2 variants to date, suggest that nemvaleukin alfa’s demonstrated ability to selectively expand CD16-positive CD56-positive NK cells while limiting the expansion of FOXP3-positive T_regs_ in patients enrolled on ARTISTRY-1 may meaningfully increase its therapeutic potential in comparison to aldesleukin and other IL-2 variants that do not exert significant differential effects on NK and T_reg_ cell populations. The safety profile across the nemvaleukin monotherapy and nemvaleukin plus pembrolizumab combination arms was generally consistent, with neutropenia as the most frequent grade 3 and grade 4 TRAEs in both arms. The neutropenia improved or resolved with limited dose modifications. Treatment discontinuations due to TRAEs occurred with frequencies of 3% and 4% in the monotherapy and combination arms, respectively [26].

Nemvaleukin alfa was evaluated as either monotherapy or in combination with pembrolizumab in the ARTISTRY-2 trial (NCT03861793) and induced selective expansion of CD8-positive T cells and NK cells to a similar or greater extent than the IV nemvaleukin dosing utilized in ARTISTRY-1. The ARTISTRY-2 trial evaluated the safety of SC nemvaleukin in phase 1 dose escalation cohorts leading to the selection of 3.0 mg SC every 7 days as the recommended phase 2 dose (RP2D). Of the 57 patients treated with SC nemvaleukin during the phase 1 dose escalation, 31 (54%) had SD at the time of the first scheduled radiographic assessment. Seventeen out of 37 (46%) patients with more than two evaluable on-treatment scans had SD on two or more consecutive scans. The R2PD of nemvaleukin alfa administered at 3.0 mg SC every 7 days in combination with IV pembrolizumab every 21 days was taken forward to the phase 2 efficacy expansion in select solid tumors including non-small cell lung cancer, head and neck squamous cell carcinoma, platinum-resistant ovarian cancer, and gastric/gastroesophageal junction adenocarcinoma [26].

## 7. Bempegaldesleukin

Another potential strategy for improving the therapeutic window of IL-2 involves biochemical modifications of recombinant IL-2. By conjugating several releasable polyethylene glycol (PEG) chains to aldesleukin to increase its half-life, bempegaldesleukin (BMS) functions as an IL-2 prodrug and a CD-122 preferential IL-2 pathway agonist. The clustering of PEG chains on lysine residues prevents bempegaldesleukin from binding to the trimeric form of the IL-2 receptor, and its activity for preferentially driving the proliferation of effector T cells increases as it is hydrolyzed.

However, data from randomized phase 3 clinical trials presented at the ESMO Congress in September 2022 showed reduced clinical efficacy of bempegaldesleukin in both advanced melanoma and advanced renal cell carcinoma when combined with a PD-1 inhibitor. The phase 3 PIVOT IO 001 trial (NCT03635983) randomized 783 patients with untreated unresectable or metastatic melanoma to receive either bempegaldesleukin at 0.006 mg/kg plus nivolumab or nivolumab alone. The primary endpoint of objective response rate (ORR) with bempegaldesleukin plus nivolumab was 27.7% vs. 36.0% with nivolumab (two-sided *p* = 0.0311) with a median follow-up of 19.3 months, and the primary endpoint of median progression-free survival (PFS) with bempegaldesleukin plus nivolumab was 4.17 months (95% CI, 3.52–5.55) vs. 4.99 months (4.14–7.82) with nivolumab (HR 1.09 with 97% CI of 0.88 to 1.35; *p* = 0.3988), with a median follow-up of 11.6 months. In addition to the lower ORR in the combination arm, grade 3 and 4 TRAEs and serious AEs were higher with bempegaldesleukin plus nivolumab (21.7% and 10.1%) vs. nivolumab (11.5% and 5.5%). An AE of special interest, ischemic cerebrovascular events, was higher with bempegaldesleukin plus nivolumab (2.6%) vs. nivolumab (0.8%), and there were three bempegaldesleukin plus nivolumab treatment-related deaths vs. one nivolumab treatment-related death [28].

The phase 3 PIVOT-09 trial randomized 623 patients with untreated advanced/metastatic clear-cell renal cell carcinoma who received bempegaldesleukin at 0.006 mg/kg IV plus nivolumab 360 mg IV every 3 weeks compared to sunitinib 50 mg orally once a day for 4 weeks followed by 2 weeks off or cabozantinib 60 mg orally once a day. At a median duration of follow-up of 15.5 months, the ORR was 23.0% for bempegaldesleukin plus nivolumab vs. 30.6% for the tyrosine kinase inhibitor (TKI) arm. OS in the International Metastatic RCC Database Consortium (IMDC) intermediate-risk and poor-risk patients with a *p*-value of 0.19 did not pass the pre-specified alpha of 0.01 at interim analysis. The median OS was 29.0 months for bempegaldesleukin plus nivolumab and was not reached for the TKI arm (HR 0.82, 99% CI: 0.56 to 1.21). In the IMDC all-risk group, the most common TRAEs (>20%) of any grade in the bempegaldesleukin plus nivolumab arm were pyrexia (32.6%), pruritus (31.3%), nausea (24.2%), eosinophilia (23.9%), hypothyroidism (22.9%), rash (22.9%), and arthralgia (20.0%). Grade ≥3 TRAEs occurred in 83 patients (26.8%), and 24 patients (7.7%) discontinued bempegaldesleukin plus nivolumab due to TRAEs. Grade 5 TRAEs occurred in 3 out of 310 patients (1.0%) [29].

## 8. THOR-707 (SAR444245)

By applying a unique synthetic biology platform that facilitates unnatural codon incorporation into proteins, a pegylated variant of IL-2 named THOR-707 (SAR444245: Synthorx) was developed, which blocks the α-binding domain while retaining high affinity for the IL-2Rβγ, hypothetically improving its therapeutic index compared to recombinant IL-2 [30]. This synthetic platform was previously used to transcribe DNA containing dNaM, an artificial nucleobase harboring the 3-methoxy-2-naphthyl group, and dTPT3 into mRNAs with two different unnatural codons and tRNAs with a cognate unnatural anticodon for the site-specific incorporation of non-canonical amino acids into a green fluorescent protein [31]. THOR-707 is currently being evaluated in several clinical trials including a phase 1/2 dose escalation and expansion study as a single agent and in combination with pembrolizumab in patients with advanced or metastatic solid tumors (HAMMER, NCT04009681). Of the 28 total patients enrolled in HAMMER, 2 patients with head and neck squamous cell carcinoma and basal cell carcinoma in the combination treatment arm and 1 patient with squamous cell carcinoma of unknown primary in the monotherapy arm achieved PR [30]. Two patients with pancreas and prostate cancer achieved SD for 9 and 6 months, respectively. The preliminary results suggest acceptable tolerability with no dose-limiting toxicity or vascular leak syndrome observed, and common AEs of flu-like symptoms (46.4%), fever (46.4%), vomiting/nausea (35.7%), and chills (32.1%) were reported. Other active phase 2 trials include evaluating THOR-707 in combination with pembrolizumab for the treatment of patients with lung cancer or mesothelioma (NCT04914897) or in combination with either pembrolizumab or cetuximab for patients with advanced or metastatic gastrointestinal cancer (NCT05104567).

## 9. TransCon IL-2 β/γ

TransCon IL-2 β/γ (Ascendis Pharma) is a long-acting prodrug that consists of an IL-2 β/γ selective agonist (developed through the permanent conjugation of a small 5 kDa methoxy polyethylene glycol (mPEG) within the IL-2Rα binding site of an IL-2 mutein) attached to a 40 kDA mPEG carrier via a transient conjugation linker. Studies in monkeys demonstrated that TransCon IL-2 β/γ has slow-release pharmacokinetics with a low peak serum concentration and a half-life greater than 30 h, substantially longer than intravenous recombinant IL-2 [32]. A first-in-human phase 1/2 dose escalation and dose expansion study (NCT05081609) is ongoing evaluating TransCon IL-2 β/γ alone or in combination with pembrolizumab, standard-of-care chemotherapy, or a TLR7/8 agonist in patients with locally advanced or metastatic solid tumors.

## 10. Future Directions

While high-dose rIL-2 demonstrated the early tantalizing prospect of leveraging cytokine-based therapies to induce complete remissions of cancers, changing the landscape for melanoma and renal cell carcinoma, its full therapeutic potential remains unrealized with the considerable limitations of low response rates, severe toxicities, and a narrow therapeutic index. With rejuvenated preclinical and clinical development of novel IL-2 variants via biochemical modifications and cytokine engineering and promising clinical data suggesting improved single-agent activity for newer agents such as nemvaleukin alfa in solid tumor malignancies beyond the established FDA indications of metastatic melanoma and renal cell carcinoma, certain IL-2 variants are poised to serve as viable platforms for combination with anti-PD-(L)1 and anti-CTLA-4 immunotherapy, chemotherapies, and targeted therapy approaches. Examples of ongoing studies with combinatory therapies include the ARTISTRY-7 phase 3 study with nemvaleukin alpha and pembrolizumab and the ABILITY phase 1/2 study with MDNA11 (an albumin-IL-2 fusion protein that preferentially activates effector T cells and NK cells), evaluated as a single agent or in combination with an immune checkpoint inhibitor in patients with advanced solid tumors (NCT05086692).

While bempegaldesleukin has yielded disappointing phase 3 clinical trial results, other IL-2 variants in clinical development that rely on different mechanisms of action hold promise for significantly expanding the therapeutic index of IL-2 with more effective induction and activation of anti-tumor cytotoxic T cells, including THOR-707 and TransCon IL-2 β/γ. Other alternatives such as immunocytokines with antigen-binding domains fused to IL-2 and immunotoxins with toxins fused to IL-2 are also being developed as distinct therapeutic options [33]. The enthusiasm for harnessing protein engineering and computationally modeled biologics is sharply reflected in these diverse examples of drug development, and it seems that the continuous exploration and interrogation of IL-2 as a viable target for cancer immunotherapy is poised to yield a new generation of promising therapeutic approaches expressly designed to mitigate immunologic toxicities.

## Figures and Tables

**Figure 1 cells-12-01316-f001:**
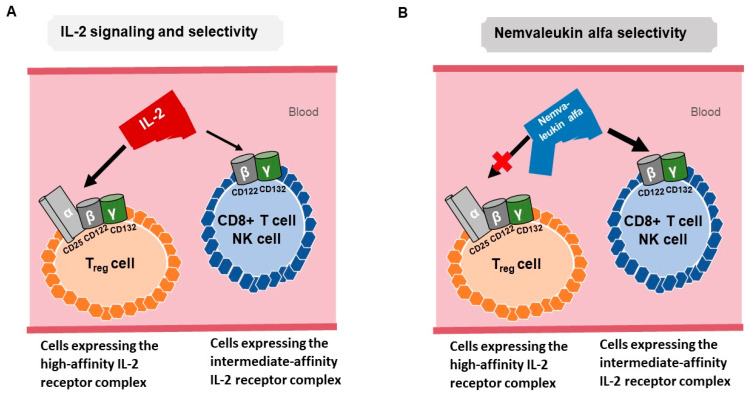
(**A**) IL-2 binds to the high-affinity IL-2 receptor complex containing α, β, and γ subunits expressed on T_reg_ and endothelial cells. This high-affinity selectivity can lead to the expansion of immunosuppressive T_regs_ and life-threatening toxicities. IL-2 also binds to the intermediate-affinity IL-2 receptor complex of β and γ subunits on CD8-positive cytotoxic T cells and NK cells. (**B**) Nemvaleukin alfa is an engineered IL-2-IL-2Rα fusion protein that selectively binds to the intermediate-affinity IL-2 receptor complex, where it preferentially activates CD8-positive cytotoxic T cells and NK cells rather than the expansion of T_regs_, enhancing the anticancer immune effects mediated by CD8-positive T cells.

**Table 1 cells-12-01316-t001:** A description of all the active and completed clinical trials for evaluating nemvaleukin alfa treatment in patients with solid tumors (https://clinicaltrials.gov/ accessed on 1 February 2023).

Title	Cancer Type	Phase of Clinical Trial and Status	Clinical Trial.gov Number
Less Frequent IV Dosing & Tumor Microenvironment (TME) Study of Nemvaleukin Alfa Monotherapy and in Combination with Pembrolizumab (ARTISTRY-3)	Advanced solid tumors	Phase 1/2, recruiting	NCT04592653
A Dose Escalation and Cohort Expansion Study of Subcutaneously-Administered Cytokine Nemvaleukin Alfa as a Single Agent and in Combination with Anti-PD-1 Antibody (Pembrolizumab) in Subjects with Select Advanced or Metastatic Solid Tumors (ARTISTRY-2)	Advanced solid tumors	Phase 1/2, active, not recruiting	NCT03861793
A Study of the Effects of Nemvaleukin Alfa on Subjects with Solid Tumors (ARTISTRY-1)	Advanced solid tumors	Phase 1/2, active, not recruiting	NCT02799095
A Study of Nemvaleukin Alfa with Pembrolizumab in Head and Neck Cancer	Non-cutaneous squamous cell carcinoma of head and neck	Phase 2, completed	NCT04144517
Nemvaleukin Alfa Monotherapy in Patients with Advanced Cutaneous Melanoma or Advanced Mucosal Melanoma (ARTISTRY-6)	Cutaneous and mucosal melanoma	Phase 2, recruiting	NCT04830124
Phase 3 Study of Nemvaleukin Alfa in Combination with Pembrolizumab in Patients with Platinum-Resistant Epithelial Ovarian Cancer (ARTISTRY-7)	Platinum-resistant ovarian cancer, fallopian tube cancer, and primary peritoneal cancer	Phase 3, recruiting	NCT05092360

## Data Availability

Not applicable.

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
