# Peer review of "Rethinking Oncologic Treatment Strategies with Interleukin-2"

_cells, 2023, doi:10.3390/cells12091316_

Round 1

Reviewer 1 Report

In this manuscript entitled "Rethinking Oncologic treatment strategies with interleukin-2", Brian Ko and colleagues provide a comprehensive summary of the clinical experience with IL-2 in oncology. The manuscript is well-written, accurate, and relevant in immuno-oncology. 

As a minor comment, the authors do not mention the relevance and clinical experience of IL-2 in the adoptive T-cell therapy setting.  This can enhance the scope of the manuscript. 

Author Response

We very much appreciate the reviewer's comment about "the relevance and clinical experience of IL-2 in the adoptive T-cell therapy setting" but elected to focusing on IL-2 as a primary systemic therapy.  Addressing how IL-2 is (or is not) used for various adoptive cell/autologous TIL therapies would be a whole new section likely covering myeloablative vs. lymphodepleting conditioning, and timing and dosing of rIL-2 in these settings.  While of clinical interest (specifically as an introduction to using nemvaleukin rather than IL-2 in this area), we felt it wasn't appropriate to inculde.  

Reviewer 2 Report

The authors review an interesting literature concerning the role of IL-2 administered as a cancer therapy from the early trials by Rosenberg et al with melanoma and renal cell carcinoma (RCC) to the present day. AS they indicate, early work was hampered/limited by capillary leak syndrome, but also by a limited efficacy in only a  subset of patients and cancers. While this may reflect a preferential binding to a high affinity receptor and thus preferential expansion of Tregs, this has by no means been a unanimously accepted explanation. It has however driven studies on a novel modified IL-2, nemvaleukin, discussed in this review, which binds to the lower affinity receptor on Teff, and is sterically blocked from binding to Tregs. Interestingly, this seems to be achieving most prominence in cancers other than melanoma and RCC, in association with checkpoint inhibitors. It would be worthwhile to consider whether this reflects an expansion in the mechanisms of cation for this reagent, and indeed whether we really understand the mechanism(s) of action of all forms of IL-2, including the PEGylated versions also discussed herein (Bempegaldesleukin;THOR-707). This in itself may lead to discussion of additional novel approaches not yet considered in a clinical context.

Author Response

We thank the reviewer for these helpful comments which we hope to have addressed on page 6 paragraph 1 of the track changes document attached. 

Reviewer 3 Report

The manuscript “Rethinking oncologic treatment strategies with interleukin-2” by Ko et al. gives an overview of biochemically modified or engineered IL-2 variants that are currently used in the clinic as immunotherapeutic approaches. The review focuses on Nemvaleukin-α (engineered IL-2-IL-2Rα fusion protein), Bempegaldesleukin (biochemical modification of rIL-2), THOR-707 (a pegylated variant of IL-2) and TransCon IL-2βγ (methoxy polyethylene glycol conjugated variant) and the clinical trials that have been performed or are currently ongoing with these drugs.

Major Concern

There is substantial overlap with a review recently been published (9 Nov 2022) in Science Translational Medicine (https://pubmed.ncbi.nlm.nih.gov/36350987).

Minor issues  

Page 1: “Since the discovery of interleukin-2 (IL-2) as a potent growth factor for human T cells in 1976 promoting T cell expansion and maintenance of function, there ……”. I recommend to change it to “Since the discovery of interleukin-2 (IL-2) in 1976 as potent growth factor promoting expansion and maintenance of function of human T cells, there.....”

Page 2: “effector and memory effector cells”. I recommend to change it to “effector memory and terminal effector cells”.

Page 2: it has not been clearly described that Tregs mainly express the high affinity trimeric receptor, whereas T cells and NK cells predominantly express the intermediate receptor.

Page 3: Cmax has not been explained. Do the authors mean peak serum concentration?

Page 4: references 24, 25: I could not find the information on checkpoint inhibitors or tyrosine kinase inhibitors in these references.

Page 4: “intratumoral live T cells and live CD8-positive T cells”. Why do the authors emphasize twice that these are live T cells?

Table 1: The name “ARTISTRY-1” has not been mentioned in the NCT trial number.

Page 5: “distinct patient cohorts of mixed tumor types”. I recommend to change it to “distinct patient cohorts with different tumor types”.

Page 7: “PO” has not been explained. Do the authors mean Per Os?

Page 7: “TKI” has not been explained. Do the authors mean Tyrosine Kinase Inhibitors?

Author Response

We thank the reviewer for very helpful comments which we have attempted to address in the attached track-changes document. 

For our response to the major concern ("There is substantial overlap with a review...")  we have would like to emphasize that our paper is more focused on providing recent clinical data for clinicians/medical oncologists and other interested parties:

  • Nemvaleukin alfa’s anti-tumor activity in mucosal melanoma, which represents a chronic unmet need, particularly when compared to the multiple therapeutic advances for cutaneous melanoma (please see page 6 paragraph 1).
  • Nemvaleukin alfa’s ongoing clinical development in cancer types outside of cutaneous melanoma and renal cell carcinoma in which rIL-2 has not demonstrated efficacy, especially platinum-resistant ovarian cancer (see page 6 paragraphs 1 & 2).
  • Highlighting bempegaldesleukin’s failure in the phase 3 PIVOT IO 001 and PIVOT-09 clinical trials, which was very concerning for actually having lower ORR when combined with nivolumab versus nivolumab monotherapy (see the bempegaldesleukin section on pages 6 & 7).
  • Alternative approaches with other pegylated IL-2 variants have been added on page 7 paragraph 3, and page 8 paragraph 1.

Minor issues are addressed in with the bullets below.

Page 1: “Since the discovery of interleukin-2 (IL-2) as a potent growth factor for human T cells in 1976 promoting T cell expansion and maintenance of function, there…”. I recommend to change it to “Since the discovery of interleukin-2 (IL-2) in 1976 as potent growth factor promoting expansion and maintenance of function of human T cells, there...”

  • Revised as suggested, page 1 paragraph 2.

Page 2: “effector and memory effector cells”. I recommend to change it to “effector memory and terminal effector cells”.

  • Revised as suggested, page 2 paragraph 2.

Page 2: it has not been clearly described that Tregs mainly express the high affinity trimeric receptor, whereas T cells and NK cells predominantly express the intermediate receptor.

  • Revised as suggested, page 2 paragraph 3.

Page 3: Cmax has not been explained. Do the authors mean peak serum concentration?

  • Changed to “peak serum concentrations”, page 3 paragraph 3.

Page 4: references 24, 25: I could not find the information on checkpoint inhibitors or tyrosine kinase inhibitors in these references.

  • Apologies; we have removed the extraneous text associated with these references, page 4 paragraph 2.

Page 4: “intratumoral live T cells and live CD8-positive T cells”. Why do the authors emphasize twice that these are live T cells?

  • Removed “live” from both, page 4 paragraph 2.

Table 1: The name “ARTISTRY-1” has not been mentioned in the NCT trial number.

  • Edited Table 1 on page 5.

Page 5: “distinct patient cohorts of mixed tumor types”. I recommend to change it to “distinct patient cohorts with different tumor types”.

  • Revised as suggested, page 5 paragraph 2.

Page 7: “PO” has not been explained. Do the authors mean Per Os?

  • Changed to “orally,” page 7 paragraph 2.

Page 7: “TKI” has not been explained. Do the authors mean Tyrosine Kinase Inhibitors?

  • Revised as suggested, page 7 paragraph 2.

Round 2

Reviewer 3 Report

The authors accurately addressed all issues